# The Aggregated Gut Viral Catalogue (AVrC): A unified resource for exploring the viral diversity of the human gut

Anastasia Galperina[1], Gabriele Andrea Lugli[2,3], Christian Milani[2,3], Willem M. De Vos[1,4], Marco Ventura[2,3], Anne Salonen[1], Bonnie Hurwitz[5,6], Alise Jany Ponsero [1,7]*

1 Human Microbiome Research Program, Faculty of Medicine, University of Helsinki, Helsinki, Finland, 2 Laboratory of Probiogenomics, Department of Chemistry, Life Sciences, and Environmental Sustainability, University of Parma, Parma, Italy, 3 Microbiome Research Hub, University of Parma, Parma, Italy, 4 Laboratory of Microbiology, Wageningen University, Wageningen, The Netherlands, 5 Department of Biosystems Engineering, The University of Arizona, Tucson, Arizona, United States of America, 6 BIO5 Institute, The University of Arizona, Tucson, Arizona, United States of America, 7 Quadram Institute Bioscience, Norwich, United Kingdom

* alise.ponsero@quadram.ac.uk

## Abstract

The growing interest in the role of the gut virome in human health and disease, has led to several recent large-scale viral catalogue projects mining human gut metagenomes each using varied computational tools and quality control criteria. Importantly, there has been to date no consistent comparison of these catalogues' quality, diversity, and overlap. In this project, we therefore systematically surveyed nine previously published human gut viral catalogues. While these catalogues collectively screened >40,000 human fecal metagenomes, 82% of the recovered 345,613 viral sequences were unique to one catalogue, highlighting limited redundancy between the ressources and suggesting the need for an aggregated resource bringing these viral sequences together. We further expanded these viral catalogues by mining 7,867 infant gut metagenomes from 12 large-scale infant studies collected in 9 different countries. From these datasets, we constructed the Aggregated Gut Viral Catalogue (AVrC), a unified modular resource containing 1,018,941 dereplicated viral sequences (449,859 species-level vOTUs). Using computational inference tools, annotations were obtained for each vOTU representative sequence quality, viral taxonomy, predicted viral lifestyle, and putative host. This project aims to facilitate the reuse of previously published viral catalogues by the research community and follows a modular framework to enable future expansions as novel data becomes available.

## Author summary

The human gut is home to a vast diversity of viruses which have been shown to play a crucial role in human health and disease. In order to explore the Human

which permits unrestricted use, distribution, and reproduction in any medium, provided the original author and source are credited.

**Data availability statement:** The data that support the findings of this study are in Zenodo DOI 10.5281/zenodo.11426065, the AVrC toolkit repository (https://github.com/apon-sero/AVrC_toolkit) and in the Supplementary material. The code used to generate each of the figures of this manuscript is available in Github https://github.com/aponsero/Figures_AVrC_manuscript.

**Funding:** This study was supported by grants the Academy of Finland (339172 to AJP), by the BBSRC Institute Strategic Programme Food Microbiome and Health BB/X011054/1 the BBSRC Core Capability Grant BB/CCG2260/1 (AJP). The funders had no role in study design, data collection and analysis, decision to publish, or preparation of the manuscript.

**Competing interests:** The authors have declared that no competing interests exist.

gut virome diversity, several research groups have mined metagenomic samples to create catalogues of viral sequences found in the human gut. In this study, we compared nine of these previously published catalogues and found that there was little overlap between them, with 82% of the viral sequences being unique to a single resource. To further expand the available data, we mined the viral content of 7,867 additional infant gut metagenomes. We then created a unified resource combining all these catalogues, called the Aggregated Gut Viral Catalogue (AVrC). The AVrC contains more than a million viral sequences from human gut metagenomes, representing nearly 450,000 different viral species. This resource, which is accessible as a relational database (10.5281/zenodo.11426064) or through a CLI toolkit (https://github.com/aponsero/AVrC_toolkit), aims to facilitate the exploration of the vast diversity of the human gut virome and its potential implications for human health.

## Introduction

Despite the increasing number of studies highlighting the importance of gut virome in health and disease, identifying viral sequences in large metagenomic datasets is still computationally challenging. Strikingly, in gut viromes, 75% to 99% of viral reads do not produce significant alignments to any known viral genome [1]. This large range can be partially explained by broad under-representation of viral sequences in most genomic databases and the overrepresentation of specific virus taxonomic groups in these databases. All in all, gut virome profiling based on viral RefSeq databases is shown to lead to a poor and incomplete delineation of the true gut virome composition [2].

Recent bioinformatic tools leverage machine learning algorithms to identify features that signal a phage origin, and typically allow for a broader recall of previously unknown sequences than reference-based approaches. VirFinder [3], DeepVirFinder [4] and Seeker [5] use a machine-learning approach to classify sequences as phage or prokaryotic based on their k-mer sequence composition. Other machine-learning tools such as VirSorter2 [6], VirMiner [7], VIBRANT [8], base their prediction on genomic features such as the relative synonymous codon usage, gene density, strand shifts, and the number of protein gene homologs. Altogether, these new approaches provide new avenues to explore the untapped viral diversity in metagenomes. However, all currently available tools are limited to the classification of assembled contigs (cannot be applied to raw sequencing reads) and typically provide a simple binary classification (viral vs non-viral).

In the past few years, several large-scale aggregation efforts mining viral sequences from metagenomes have been released. The IMG/vr database maintained by the Department of Energy (DOE) [9,10] mines both environmental and human-associated metagenomes, while several viral catalogues focus specifically on the human gut: Gregory et al. analyzed 2,697 human gut metagenomes, recovered 33,242 species-like viral operational taxonomic units (vOTUs), and

established the gut virome database (GVD) [11]. Tisza et al. collected 5,996 metagenome datasets from human gut, skin, saliva and vagina to build the Cenote human virome database (CHVD) containing 45,033 species-like vOTUs [12]. Benler et al. mined the human gut metagenomes available in NCBI in 2019 and retrieved 3738 complete phage genomes [13], Nishijima et al. explored the viral diversity associated to the gut microbiota of healthy and diseased adult Japanese [14] and Van Espen et al. surveyed the gut virome associated with 91 healthy Danish children, adolescents, and adults [15]. Nayfach et al. constructed the metagenomic gut virus (MGV) catalogue from 11,810 human gut metagenomes and retrieved 54,118 species-like vOTUs [16]. To date, the largest effort is the mining of 28,060 human gut metagenomes by Camarillo-Guerrero et al. to generate the gut phage database (GPD), which contained 142,809 species-like vOTUs [17]. Recently, some efforts focused on infant gut metagenomes, for example the COPSAC virome dataset leveraged a dataset of 465 infant fecal metagenomes and retrieved 10,021 vOTUs [18]. Despite the viral diversity of infant associated viromes [18] and the potential biological importance of the viral community in the infant gut microbiota acquisition and maturation [19,20], infant associated viruses have been largely under-explored in large-scale viral mining efforts.

Large-scale viral mining efforts in human gut microbiome have been enabled by the development of viral mining computation tools and algorithms [3,6,8,21], each with their own strength and limitation [22,23]. Despite the value of large-scale viral mining efforts in the human gut virome, there is currently no consistent comparison of the quality, diversity, and overlap of the human gut viral catalogues. Additionally, there is currently an unmet need to aggregate these viral sequences in a cohesive resource, that users can leverage to compare their own viral sequences and that can be expanded as novel viral sequences are published. Here, we aimed to [1] survey previous viral mining efforts in human fecal metagenomes, assess their quality, diversity and overlap; to [2] leverage several large scale infant cohorts to explore the viral diversity associated to infant gut microbiome; and to [3] harmonize and aggregate all currently available gut viral catalogues in a unified resource that users can leverage to compare new putative viral sequences to previous mining efforts.

## Results

### 1. Previous efforts to generate human gut viral catalogues have disparate quality and a limited overlap

We identified eight studies all aiming to generate a catalogue of viral sequences derived from human gut metagenomes, published between 2020 and 2023 (Table 1). Additionally, the IMG/Vr database is a dedicated resource developed specifically to retrieve viral sequences from metagenomes made available through the DOE JGI service and contains a number of human gut derived viral sequences. The tools and pipelines used for the identification and retrieval of viral sequences were highly different across studies. Almost all used one or several dedicated viral identification tools that perform a classification based on k-mer composition (VirFinder, DeepVirFinder, Seeker) or on the identification of genomic features (VirSorter, ViralVerify, geNomad). Only the COPSAC infant study used a completely custom viral identification method. Most studies confirmed the viral origin and evaluated the quality of the retrieved sequences using the dedicated tool CheckV (Table 1). We retrieved the list of sample and sequences included in each catalogue from their respective supplemental material and manually curated metadata information about sample country of origin, health status associated with each individual sample from NCBI and ENA with full details provided in S1 File. Although sample-associated metadata could not be retrieved for all the published studies, we estimate that altogether, more than 30,000 human fecal samples from 40 different countries were screened to generate these eight catalogues. Importantly, we estimate that 30% of the metagenomes previously screened for viral sequences were generated from healthy adult stool samples, and that at least 30% of the samples were mined in more than one catalogue (S1 File). Most catalogues screened bulk fecal metagenomes, and only the COPSAC and the DEVoC exclusively included viromes from VLP-enriched fecal samples. The GVD screened metagenomes of both VLP-enriched and bulk metagenomes.

**Table 1. List of public viral catalogues from human gut metagenomes and computational method used to generate the catalogues.**

| Source name | Source type | Reference | Viral Identification tool | QC |
|---|---|---|---|---|
| *IMG/Vr* | Metagenomic database | [10] | geNomad | CheckV |
| *Gut virome database (GVD)* | Metagenomic catalogue of human gut virome | [11] | VirSorter | BUSCO hmmsearch |
| | | | VirFinder | |
| | | | CAT | |
| *Gut Phage database (GPD)* | Metagenomic catalogue of human gut virome | [17] | VirSorter | Custom ML model CheckV |
| | | | VirFinder | |
| *Metagenomic gut virus (MGV)* | Metagenomic catalogue of human gut virome | [16] | Custom protein search | CheckV |
| | | | VirFinder | |
| | | | VirSorter | |
| *Cenote human virome database (CHVD)* | Metagenomes from adult gut, skin, vaginal and oral. | [12] | Cenote-Taker 2 | CheckV |
| *COPSAC infant phages* | Metagenomes from infant from the COPSAC cohort | [18] | Protein based custom identification | Custom QC |
| *Gut Phages (KGP)* | Metagenomes from adult gut virome | [13] | Protein based identification Seeker | Custom QC |
| | | | ViralVerify | |
| *Japanese 4D catalogue* | Metagenomes from adult gut virome | [14] | DeepVirFinder | CheckV |
| | | | HMM-HMM | |
| | | | fetchMG | |
| | | | barrnap | |
| | | | DIAMOND | |
| *Danish Enteric Virome Catalog (DEVoC)* | Metagenomes from children, adolescents, and adults | [15] | VirSorter | CheckV |
| | | | Database search | |
| | | | Gene content and structure | |

While all pipelines used to generate each catalogue contained a quality control step, the overall quality of the sequences as evaluated by CheckV where highly variable with catalogues such as DEVoC composed of more than 75% of low-quality sequences, while others, such as the MGV, KGP, or Japanese 4D, containing less than 5% of low-quality sequences (Fig 1A). As some viral identification tools can be biased toward a misclassification of plasmids as viral sequences, we assessed the potential plasmid contamination of each catalogue using geNomad. Most catalogues contained some plasmid contamination, with the GVD containing more than 10% of sequences potentially arising from plasmids. Notably, the recent release of the IMG/Vr database includes a plasmid decontamination step, and therefore did not contain any detectable plasmid contamination (Fig 1A).

We next evaluated the overall overlap between all published catalogues, dereplicating the 345,613 sequences from all catalogues into 239,298 species-like vOTUs. Strikingly, 82% of vOTUs (n = 195,153), were present in only one of the catalogues, suggesting low redundancy within these catalogues and justifying the need for a unified resource. The MGV and GPD were found to have the largest overlap in absolute number of sequences (n = 15,453), which can be explained by the large size of the two resources, and the overlap in the samples included in these two mining efforts (Fig 1B). In proportion of sequences, most catalogues contained more than 50% unique sequences that were not found in any other published catalogues, except for the IMG/Vr Gut subset (25% of unique vOTUs) and the KGP (28%). Surprisingly, only 38 vOTUs appeared in six or more of the studies, with one found in all eight resources containing adult samples. All 38 vOTUs are bacteriophages

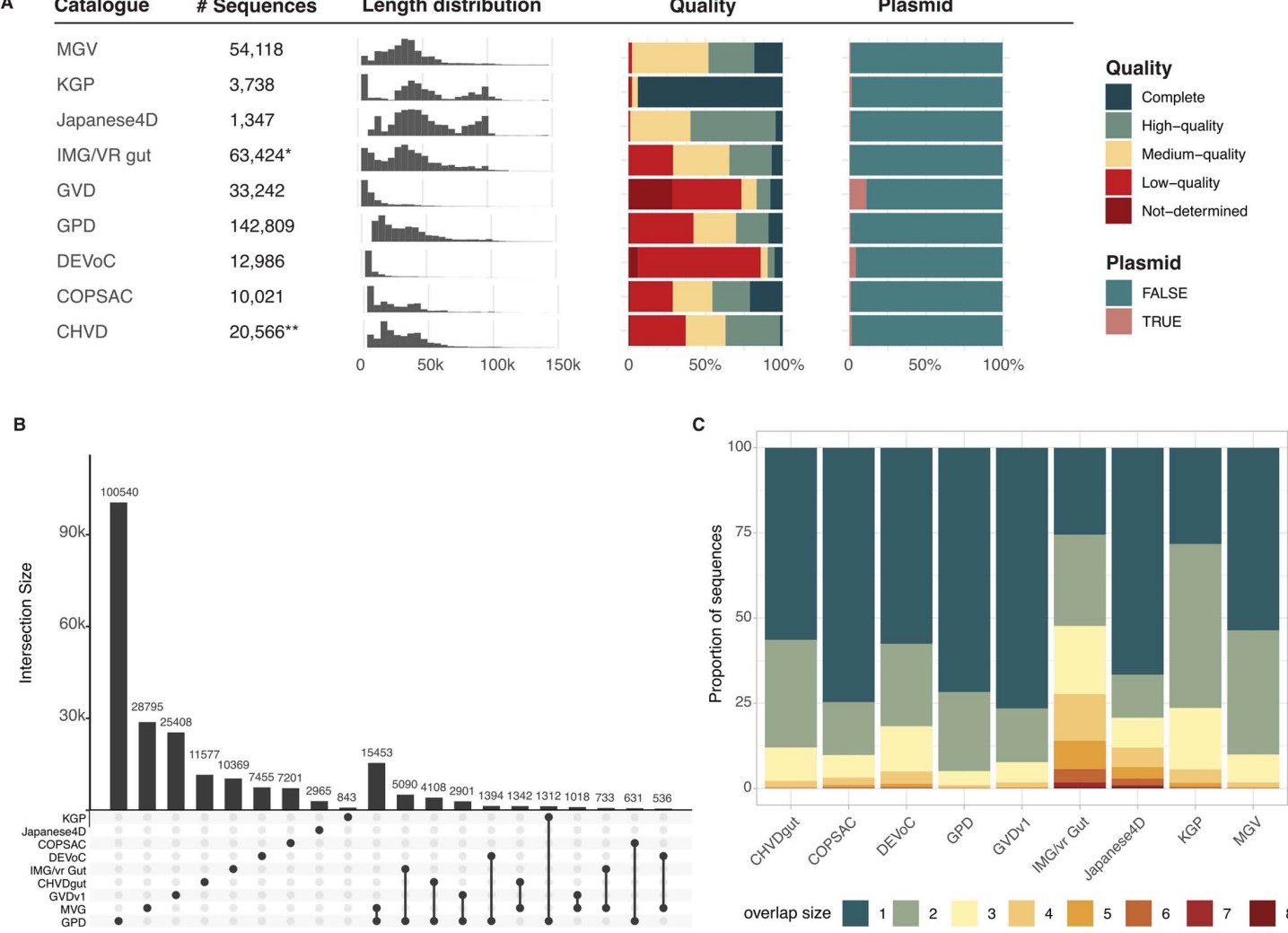

**Fig 1. Overview of previously published gut viral catalogues.** A: Previously published catalogues size, viral sequence length distribution, viral sequences quality and potential plasmid contamination.B: UpSet plot of the vOTU overlap between the previously published catalogues. The viral sequences from all the catalogues were clustered into vOTU and shared vOTU are defined as a cluster that grouped sequences from different catalogues. The intersection size was computed as the number of vOTU shared by the catalogues. The columns are sorted based on the vOTU counts per catalogue and their overlap between all combinations of catalogues. C: Proportion of unique and shared vOTU in the previously published catalogues. The sequences in the catalogues were clustered into vOTU and the "overlap size" of each vOTU was defined as the number of catalogues that contained at least one sequence for that vOTU. An overlap size of one signifies that the vOTU was uniquely found in the considered catalogue. * Subset of vOTUs from "Human gut" ecosystem accessed July 2023. ** Subset of vOTUs signaled as present in gut metagenomes.

belonging to the *Caudoviricetes* Class, potentially infecting genera such as *Bacteroides*, *Bifidobacterium* and *Phocaeicola*. Importantly, 29 of them were found to be integrated in their host genomes, giving a potential explanation for their ubiquitous presence in the catalogues. The characteristics of these high prevalence vOTUs are available in S2 File.

## 2. Screening viral sequences from more than 7,000 infant gut metagenomes

To complement the large proportion of healthy adult gut metagenomes previously screened in the published catalogues, we selected 12 large-scale infant studies, with fecal samples collected between birth and two years of age from nine different countries (S1 File). Notably, the largest collection of 2328 samples that derived from the HELMi (Health and

Early life Microbiota) cohort dataset (PRJEB70237) also included samples from the infant's parents [24]. A total of 7,867 fecal metagenomes were assembled and screened for viral sequences. We retrieved 1,205,739 putative viral sequences, among which were 44,525 high quality and 8,360 complete viral genomes (Fig 2). It is important to note that our screen used a particularly lenient threshold of viral quality control compared to previous viral mining efforts, in order to allow for a broader retrieval of the viral diversity as additional quality filters were used when merging these putative viral sequences with the other viral resources in the aggregated viral catalogue. These sequences were clustered into 648,848 species-like vOTUs, and as expected, 78% of these vOTUs were singletons, once again highlighting the high viral diversity associated with the human gut.

### 3. Building a modular and reusable unified human gut viral resource

From the previously published mining efforts and the additional infant viral sequence collection, we built a unified resource called the Aggregated Gut Viral Catalogue (AVrC). After clustering the viral sequences retrieved from the eight published catalogues, the gut subset of the IMG/Vr and the viral sequences retrieved from our infant fecal metagenome screening, we selected vOTUs for which the representative sequence was longer than 5000 bp or was annotated as "high-quality" or "Complete" by CheckV. In this resource, a minimum size of 5kb was selected to limit the potential false positives viral classification of short sequences [23]. Each putative viral sequence of the catalogue was annotated for sequence quality, potential plasmid contamination, predicted viral taxonomy, predicted viral lifestyle and putative host. The first release of the AVrC contains a total of 1,018,941 unique sequences clustered into 449,859 vOTUs, with 8% complete (n = 36,802), 21% high (n = 93,290) and 22% medium (n = 98,374) quality representative sequences, the rest of the sequences classified by CheckV as low quality or could not be assessed by the tool (Fig 3A). Importantly, most vOTUs of the AVrC are singleton (65% of the 449,859 vOTUs, n = 294,300), and the vOTU accumulation plot suggests that despite the large-scale efforts in mining the human gut virome, the total species-level viral diversity has not yet been captured (Fig 3B). Only 67,081 vOTUs of the AVrC (15% of the 449,859 vOTUs) clustered at least one sequence from the previously published catalogue and a viral sequence from our infant microbiome mining effort.

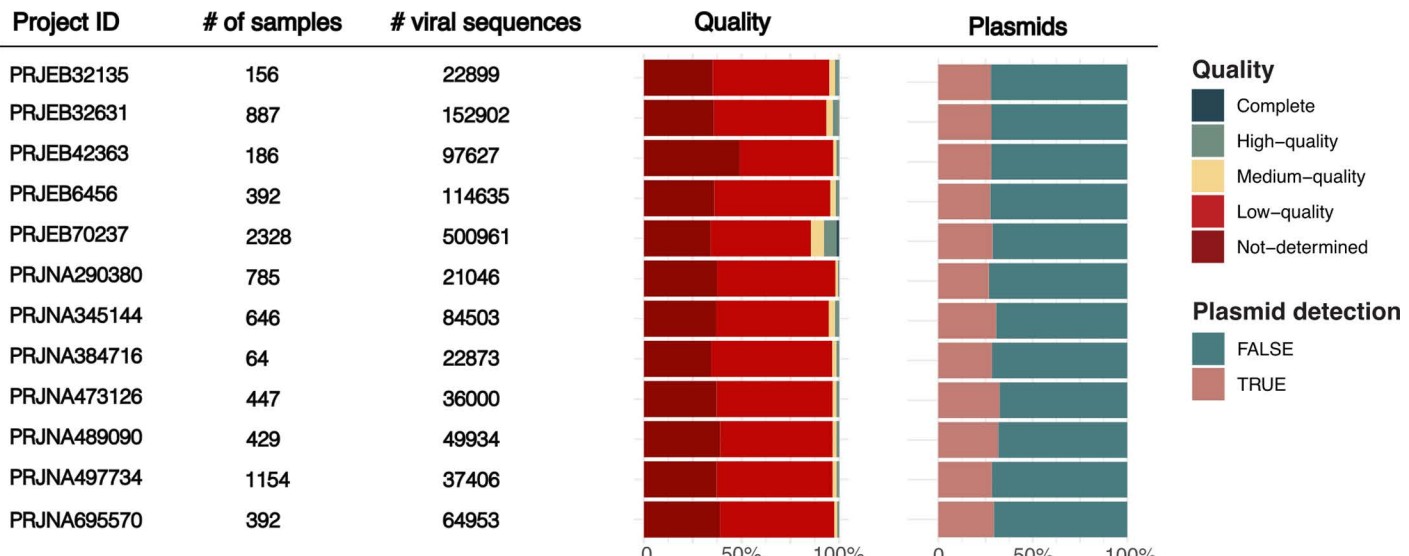

**Fig 2. Viral screening of more than 7,000 infant fecal metagenomes.** Overview for each infant project of the number of samples, number of putative viral sequences retrieved and their quality as well as the potential plasmid contamination.

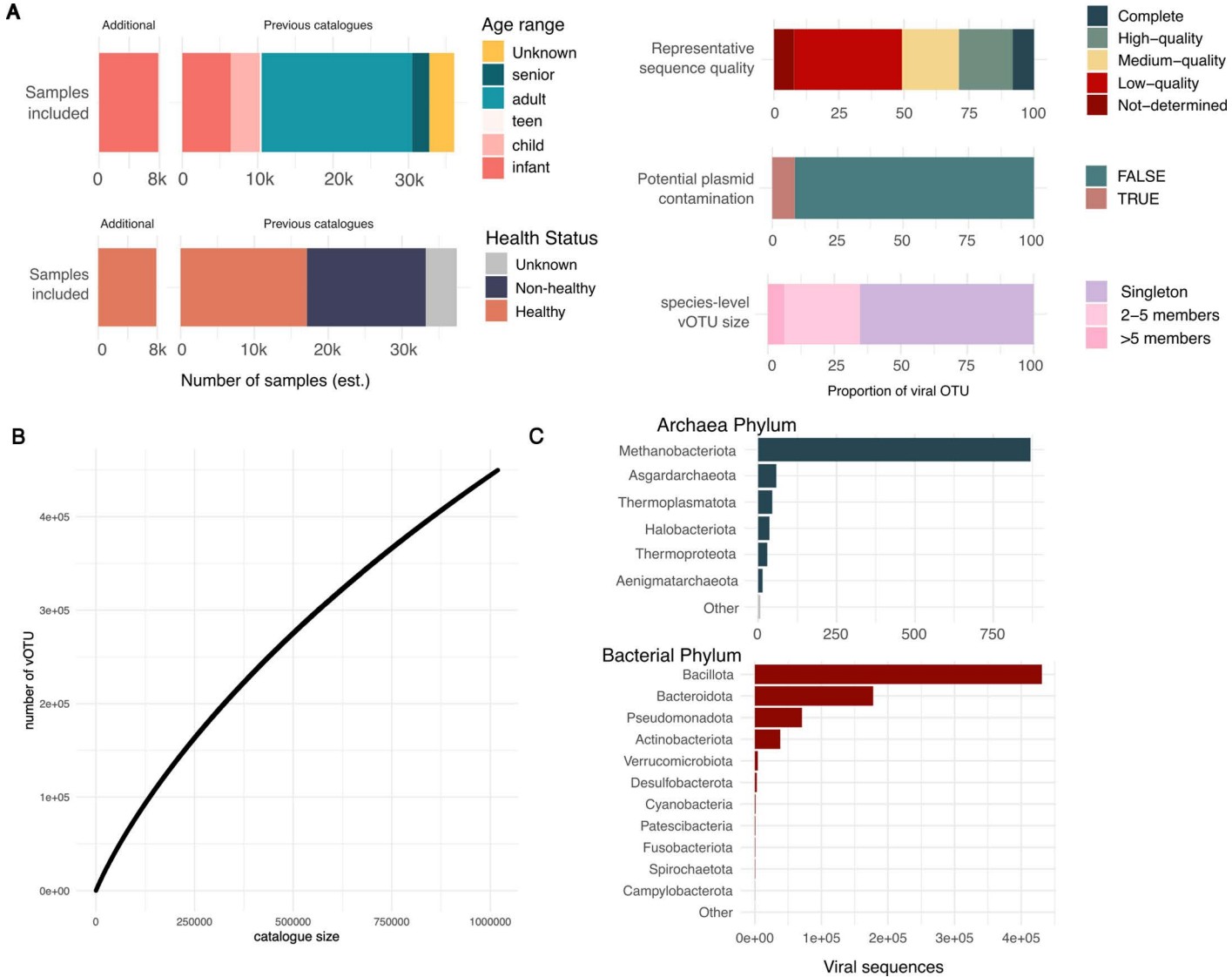

**Fig 3. Aggregated Viral Catalogue (AVrC) overview.** A: Schematic overview of the AVrC construction. The AVrC included 9 previously published catalogues and resources [10–18] and more than 7,000 additional infant gut metagenomes (PRJEB70237, PRJNA345144, PRJEB32135, PRJEB6456, PRJNA384716, PRJNA473126, PRJNA290380, PRJEB42363, PRJNA695570, PRJEB32631, PRJNA497734, PRJNA489090). The metadata for age and health status associated to previously published catalogues were extracted and manually curated when possible (excluding the IMG/Vr dataset and the KGP). An estimation of the mined sample counts per age group and health status were computed. For each vOTU, the representative sequence quality was assessed using CheckV and the potential plasmid contamination was assessed using geNomad. The vOTU size was calculated as the number of sequences grouped into a single cluster by mmSeqs2. B: Accumulation curves of the AVrC at the species-level vOTU. C: Predicted host phylum distribution for the viral sequences contained in the AVrC. The putative host for each viral sequence was obtained from iPHoP. Sequences without any predicted putative host are not displayed in the figure.

As expected, the three most abundant viral classes retrieved are the bacteriophage classes with *Caudoviricetes* constituting 83% (n = 373,751), *Malgrandaviricetes* making 2% (n = 8,083) and *Faserviricetes* being 0.1% (n = 847) of the catalogue's vOTUs. We estimated that the AVrC contains 58% of temperate bacteriophages (n = 263,079), and 34% of virulent bacteriophages (n = 153,738), with the remainder vOTUs lifestyle being uncertain. Using iPhop, 305,179 (68% of the total

449,859 AVrC vOTUs) of the vOTUs could be associated to at least one predicted prokaryotic host, and the predicted hosts for these viral species were part of the Bacillota (40% of the total 449,859 AVrC vOTUs n = 179,535), Bacteroidota (14%, n = 61,043); Pseudomonadota (5%, n = 23,173) and Actinobacteriota (4%; n = 16,795) phylum, corresponding to the major bacterial taxa found in the human gut (Fig 3C). The number of vOTUs retrieved per phylum was strongly correlated to the number of predicted host genera per phylum (Spearman correlation, p < 2.2e-16, rho = 0.96), suggesting that the taxonomic diversity of these phylum in the gut could be driving the high representation of vOTUs infecting these taxa in the AVrC. Interestingly, the catalogue also included phages infecting less common prokaryotic groups; as an example, the AVrC contains 393 vOTUs predicted to infect Spirochaetota species found exclusively in non-industrialized population [25] and contains 610 vOTUs predicted to infect archaeal species.

Importantly, the AVrC was implemented as a modular relational database to ensure easy additions or updates of the datasets and annotations. Indeed, as new catalogues and new bioinformatic tools are updated and published, we anticipate a need to continuously update the resource. The database is composed of sequence files in Fasta format containing all the sequences or the subset of representative sequences of each vOTU. All sequences can be easily linked to their individual annotations by five different tools (CheckV [26], geNomad [21], PhaGCN [27], PhaTyp [28] and iPHoP [29]). Finally, these tool annotations were combined to generate three global annotation tables that summarize the sequence quality, viral taxonomy and lifestyle, and host information (Fig 4A).

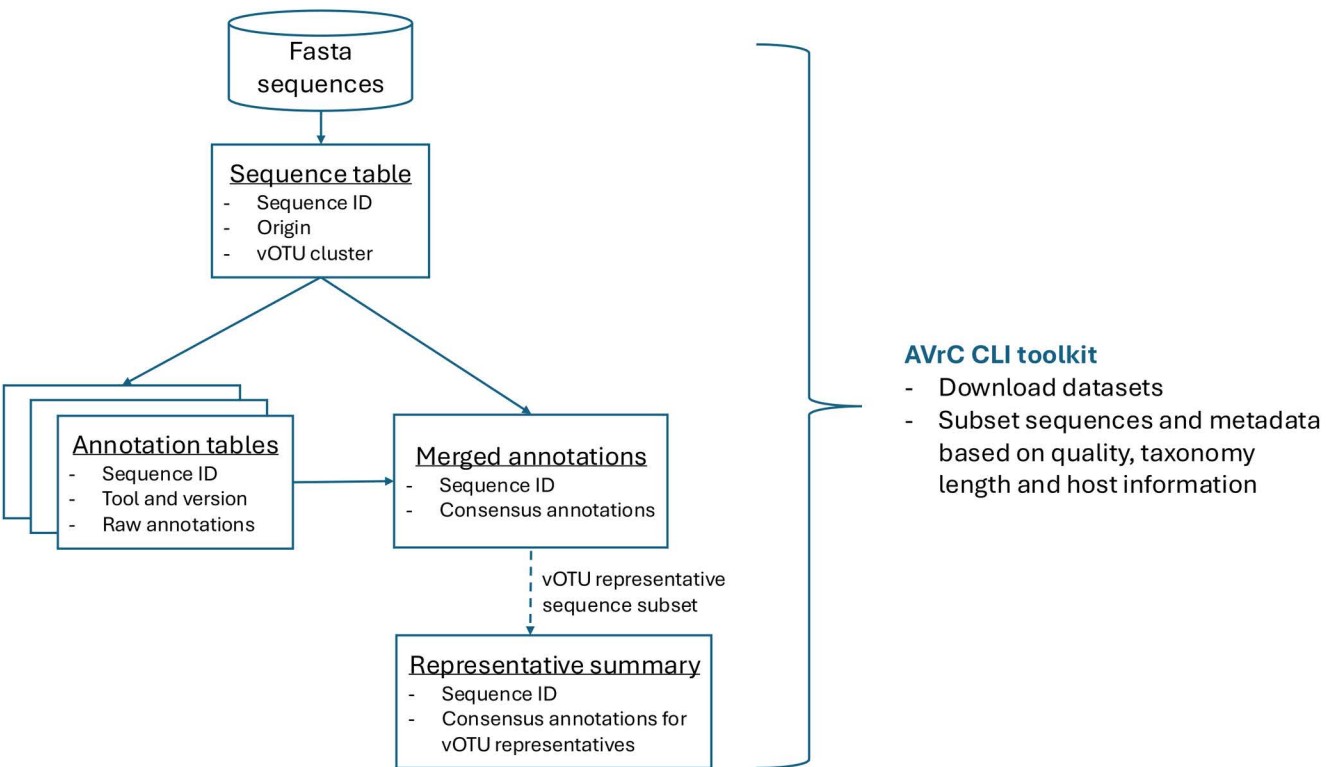

**Fig 4. Aggregated Viral Catalogue (AVrC) structure and interface AVrC database schematical structure.** The AVrC database contains a fasta sequence catalogue containing the viral sequences in a Fasta format. The annotations of the sequences are grouped in three types of tables: [1] the raw output of each annotation tools, [2] the merged and harmonized annotations recapitulating the information concerning the sequence's quality, taxonomy lifestyles and the predicted host information, and [3] a summary table containing the merged information for the vOTU representative sequences. The database is made available as csv files and a relational sql database in Zenodo (https://doi.org/10.5281/zenodo.11426064) This summary table is searchable through the AVrC toolkit, allowing users to select and search and select subsets of the dataset (https://github.com/aponsero/AVrC_toolkit).

To enable a greater reusability of the AVrC, the viral sequence catalogue and the viral annotations are available as a Zenodo archive and through the AVrC CLI toolkit. The AVrC toolkit enables users to efficiently select and retrieve subsets of the AVrC catalogue through a user-friendly CLI interface. The AVrC toolkit's download function allows to fetch the AVrC sequences files and annotations, and the filter function allows users to subset the vOTU representative sequences according to their sequence quality, length, viral taxonomy and putative hosts (Fig 4B). Data subsets of interest are also provided for an easy direct download of the sequences and annotations, including a subset containing only high-quality vOTU representative sequences and the subset of all bacteriophage sequences.

## Discussion

Recently, several large-scale efforts have been made to better understand and characterize the human gut virome, in particular by mining human gut metagenomes to generate viral catalogues. These resources are critical to enable the description of the diversity of viruses in human associated ecosystems and constitute valuable resources for the research community. Typically, each of these studies leveraged different tools, methods, and applied different quality control criteria, making the direct comparison of these resources impossible.

Here, we surveyed eight previously published studies and the subset of human gut viral sequences from the IMG/Vr database [10–18]. As expected, a large proportion of the fecal metagenomes surveyed by these catalogues were collected from healthy adults from western countries, reflecting the current known bias in human metagenome sampling. When assessing the sequence quality of these previously published studies, we observed the consistent presence of potential plasmid sequences in the datasets, in particular for the oldest resources. We therefore suggest that future viral mining efforts include a quality control step for potential plasmid contamination [10]. Strikingly, most of the viral sequences were found to be unique to one catalogue, despite the overlap between metagenomes mined by the different catalogues. This result could be explained by the impact of the computational viral detection methods on the viral sequences retrieved from these metagenomes. This result is consistent with previous observations suggesting the large impact of the chosen viral mining tool and computational approaches on the viral community composition retrieved [23].

To complement the previous mining efforts, we screened an additional 7,867 fecal infant gut metagenomes for viral sequences, retrieving over 1.2 million putative viral sequences that were clustered into 648,848 species-level viral operational taxonomic units (vOTUs). These infant viral sequences were combined with previously published human gut viral catalogues to construct the Aggregated Gut Viral Catalogue (AVrC) - a unified modular resource containing 1,018,941 dereplicated viral sequences clustered into 449,859 vOTUs with a representative sequence longer than 5kb. The large proportion of vOTUs belonging to the Caudoviricetes class reflects their central role in gut microbiome dynamics, particularly through their interactions with key bacterial phyla like Bacteroidota and Bacillota. Importantly, this also highlights the current need to explore lesser-studied viral group, that are currently biased against by most viral mining tool [23]. The identification of 393 vOTUs targeting Spirochaetota, a bacterial taxon found in non-industrialized populations, suggests that gut viromes may adapt to distinct microbial communities shaped by lifestyle and diet, and highlights the current gap in understanding of gut viromes from non-westernized population. The high proportion of predicted temperate vOTUs in the AVrC confirm the importance of lysogeny in the human gut environment and the need to also capture the prophage diversity associated to the human gut [30,31]. Despite these large-scale mining efforts, our clustering results suggest that the species-level viral diversity in the human gut has not been completely captured. Future efforts to capture the full diversity of human gut viruses should combine the harmonization of large-scale gut viral catalogues, the exploration of under-represented populations, the integration of long-read sequencing technologies, and should be able to adapt to the novel developments in both computational and experimental approaches.

The AVrC was constructed as a modular relational database that provides extensive annotations of the sequences included in the catalogue, including the sequence quality and potential plasmid contamination, viral taxonomy, predicted lifestyle, and putative microbial hosts. The AVrC is available through a CLI toolkit to enable easier customized

querying and retrieval of sequences/annotations. In the future, we aim to expand the AVrC through the continued integration of novel catalogues recently published. As an example, a novel catalogue surveying infant fecal viral diversity [32], and a virome catalogue of a large colorectal cancer screening program [33], were published during the redaction of this manuscript and will be integrated in the upcoming version 2 release of the AVrC. Additionally, recent reports showed the potential of long-read sequencing technology and hybrid approaches in the exploration of the human gut virome [34]. Comparisons of the AVrC with culture-based viral genomic resources such as the Inphared database [35] would allow to explore the diversity of phages currently not captured by viral culture approaches. Finally, as computational methods to annotate and explore the viral diversity are in constant improvement, the AVrC was built to rapidly allow for the addition of novel annotations and to facilitate the update of the annotations when new reference databases are published.

Importantly, at the time of writing, other viral catalogue unification efforts are underway, in particular the Unified Human gut Virome Catalog (UHGV, available at https://github.com/snayfach/UHGV) that draw from similar data sources as the AVrC. Indeed, the UHGV focuses on providing a curated, high-quality reference dataset through stringent quality filtering and state of the art annotation pipelines and currently includes 168,570 vOTUs. In contrast, AVrC is designed as a comprehensive aggregation resource that preserves access to varying quality levels of viral sequences, allowing researchers to apply their own quality thresholds as needed. AVrC's modular architecture enables users to select sequence subsets based on multiple quality metrics, access both complete and partial viral genomes and apply custom filtering criteria through a dedicated command-line interface. The aim of the AVrC is to easily integrate new viral sequences as they become available, as we believe that this flexibility is particularly valuable for research questions where non-complete viral sequences or lower-quality predictions might still provide valuable insights. Importantly, both approaches are addressing a current need to harmonize large scale viral catalogues into a consistent dataset.

The overview of previously published mining efforts highlighted several gaps and biases in our current understanding of the human gut virome. First, the geographical bias towards adult western populations can be addressed by actively incorporating mining from underrepresented age groups and regions. While in this first version of the AVrC we have expanded coverage by including infant samples from 9 different countries, the viral diversity from certain geographic regions and demographic groups remains underrepresented, and results should be interpreted accordingly when analysing samples from these populations. Second, the impact of the computational viral detection tools is mitigated in the AVrC as the resource combines and allows to compare multiple detection methods, reducing tool-specific biases. The modular nature of the AVrC will facilitate building upon these improvements, as new datasets and new detection or viral annotation methods can be integrated in the resource while maintaining backwards compatibility.

## Methods

### Dada source: Published gut viral catalogues and infant gut metagenomes viral mining

Literature search for previously published human gut viral mining efforts from PubMed was performed in 2023 and allowed the identification of eight relevant studies (**Table 1**). The IMG/Vr and CHVD catalogues were filtered to retrieve only viral sequences obtained from human fecal metagenomes. The original sequence names were mapped to a unified naming convention across the datasets for easier integration in the AVrC. The mapping between the original and new sequence names is available in the Zenodo archive and on the AVrC website.

We additionally selected 12 gut metagenome projects from large-scale infant birth cohorts and downloaded the metagenomes directly from NCBI or ENA (PRJEB70237, PRJNA345144, PRJEB32135, PRJEB6456, PRJNA384716, PRJNA473126, PRJNA290380, PRJEB42363, PRJNA695570, PRJEB32631, PRJNA497734, PRJNA489090). Quality controlled reads were assembled using Megahit v1.2.9 [36] for PRJEB70237 and the METAnnotatorX2 pipeline [37] using

Spades v3.15 [38] (other projects). Assembled contigs longer than 500 bp and with a coverage above 5x were classified as viral or non-viral using DeepVirFinder v1.0 [4] and VirSorter2 v2.2.3 [6]. Putative viral sequences were defined as follows: DeepVirFinder score above 0.9 or VirSorter2 viral/prophage classification. The putative viral contigs were further confirmed using CheckV v0.8.1 [26] and contigs longer than 1kb with no detected viral genes and at least one cellular gene was discarded. The sequences were dereplicated using MMseqs2 [39] with 99% identity over 90% of shortest sequences.

**Viral sequence clustering and annotations.** The putative viral sequences were clustered into species-like vOTUs using MMseqs2 [39] with a 95% identity over 75% of shortest sequences as commonly used [40]. The longest sequence for each cluster was chosen as a vOTU representative. Finally, the vOTUs with a representative sequence above 5,000 bp or classified as "high-quality" or "Complete" by CheckV were selected and kept in the AVrC.

All sequences were annotated by geNomad v1.7.4 [21] and PhaGCN [27] to obtain high level viral classification, following the new ICTV convention. GeNomad was also used to identify potential plasmid sequences. The putative viral lifestyle strategy was determined using PhaTyp [28] as well as the annotations derived from CheckV and geNomad. Briefly, four categories were generated: temperate (PhaTyp classifies the sequence as temperate with a score of>= 0.7 or CheckV and geNomad predict a prophage sequence), uncertain temperate (PhaTyp classifies the sequence as temperate with a score of < 0.7), virulent (PhaTyp classifies the sequence as virulent with a score of>= 0.7) and uncertain virulent (PhaTyp classifies the sequence as virulent with a score of < 0.7). Host prediction for the viral sequences was obtained using iPHOP v1.3.3 [29], a tool that leverages six distinct methods including both host-based tools (e.g., CRISPR markers, prophage in host genome, etc.) and phage-based tools (e.g., alignment with phages with known hosts) and merges their results to provide the user with a candidate host genus for each viral sequence. An overview of the workflow is summarized in S1 Fig.

### AVrC database and interface

The AVrC sequence catalogue and annotations are available in fasta and csv format in a Zenodo archive (10.5281/zenodo.11426065). The AVrC toolkit is made available in Github (https://github.com/aponsero/AVrC_toolkit). This CLI toolkit a python package working with python >3.8 and which requires the prior installation of seqkit. The installation and usage along with a tutorial is provided in the toolkit documentation on Github Wiki (https://github.com/aponsero/AVrC_toolkit/wiki).

### Supporting information

**S1 File. Description of the samples screened by the gut viral catalogues.**
(XLSX)

**S2 File. Description of the high prevalence vOTUs.**
(XLSX)

**S1 Fig. Computational workflow overview.**
(TIFF)

### Acknowledgments

We thank the Finnish IT Centre for Science and the NBI Research computing for providing the computational resources used for this project. We thank the other group members of the Microbes inside lab for their helpful contributions, in particular Roosa Jokela and Dollwin Matharu. We thank Prof. Kaija-Leena Kolho for helpful discussions. Finally, we thank Dr. Andrea Telatin and Dr. Ryan Cook for their support in finalizing this study and the QIB core bioinformatics team.

## Author contributions

**Conceptualization:** Anastasia Galperina, Gabriele Andrea Lugli, Christian Milani, Willem M. De Vos, Marco Ventura, Anne Salonen, Bonnie Hurwitz, Alise Jany Ponsero.

**Data curation:** Anastasia Galperina, Gabriele Andrea Lugli, Alise Jany Ponsero.

**Formal analysis:** Anastasia Galperina, Alise Jany Ponsero.

**Funding acquisition:** Alise Jany Ponsero.

**Investigation:** Gabriele Andrea Lugli, Alise Jany Ponsero.

**Methodology:** Anastasia Galperina, Gabriele Andrea Lugli, Christian Milani, Marco Ventura, Anne Salonen, Alise Jany Ponsero.

**Project administration:** Alise Jany Ponsero.

**Resources:** Anne Salonen.

**Supervision:** Willem M. De Vos, Anne Salonen, Bonnie Hurwitz, Alise Jany Ponsero.

**Visualization:** Anastasia Galperina, Alise Jany Ponsero.

**Writing – original draft:** Anastasia Galperina, Alise Jany Ponsero.

**Writing – review & editing:** Anastasia Galperina, Gabriele Andrea Lugli, Christian Milani, Willem M. De Vos, Marco Ventura, Anne Salonen, Bonnie Hurwitz, Alise Jany Ponsero.

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
