## [Decision Letter · Decision Letter 0]

22 Jan 2025

PCOMPBIOL-D-24-01049

The Aggregated Gut Viral Catalogue (AVrC): A Unified Resource for Exploring the Viral Diversity of the Human Gut

PLOS Computational Biology

Dear Dr. Ponsero,

Thank you for submitting your manuscript to PLOS Computational Biology. After careful consideration, we feel that it has merit but does not fully meet PLOS Computational Biology's publication criteria as it currently stands. Therefore, we invite you to submit a revised version of the manuscript that addresses the points raised during the review process.

Please submit your revised manuscript within 60 days Mar 24 2025 11:59PM. If you will need more time than this to complete your revisions, please reply to this message or contact the journal office at ploscompbiol@plos.org. Please include the following items when submitting your revised manuscript:

We look forward to receiving your revised manuscript.

Kind regards,

Iddo Friedberg, Ph.D.

Academic Editor

PLOS Computational Biology

Rob De Boer

Section Editor

PLOS Computational Biology

**Journal Requirements:**

3) Please ensure that the funders and grant numbers match between the Financial Disclosure field and the Funding Information tab in your submission form. Note that the funders must be provided in the same order in both places as well.

**Reviewers' comments:**

Reviewer's Responses to Questions

**Comments to the Authors:**

Reviewer #1: This article, a valuable tool for analyzing human gut microbiome viromes, is proof of the authors' remarkable effort. We highly appreciate their work in classifying, validating, and annotating the sequences obtained for the human gastrointestinal tract virome.

However, to be accepted, it needs to review a series of comments in the attached file.

I will be honored to review the revised version of this manuscript to proceed with its possible publication in the journal PLOS Computational Biology.

Reviewer #2: Dear authors,

The manuscript titled "The Aggregated Gut Viral Catalogue (AVrC): A Unified Resource for Exploring the Viral Diversity of the Human Gut" presents a significant effort to unify and enhance resources for studying the gut virome. The authors have systematically analyzed existing viral sequence catalogues and integrated new datasets to create a more comprehensive database for studying the virome in the human gut. The study is well-structured, and presents novel insights. There are areas where the manuscript could be improved for clarity, methodological rigor, and presentation. Please find below some comments to be addressed.

Introduction:

- Line 86: Quality assessments and diversity analyses were presented throughout the manuscript but there weren't any measures on completeness explicitly. Is your measure of completeness based solely on overlaps between databases? Consider clarifying or revising this statement.

Results

- Lines 107-111: Briefly describe your method for estimating the number of fecal samples, countries, sample screening, health status, and overlaps in catalogues.

- Line 148: Specify which catalogue contains 82% of the unique sequences.

- Lines 149-151: The explanation regarding the relationship between resource size and shared sequences is unclear. MGV is not the largest catalogue. Shouldn’t IMG/VR and GPD, being larger, have higher shared sequences? Based on Fig. 1C, IMG/VR and KGP show higher shared sequences, while GPD (142K sequences) shows fewer. Clarify this reasoning.

- Line 167: Figure 2 only has one panel; remove "A"

- Line 182: Justify the inclusion criterion for vOTUs (>5000bp). Can shorter sequences still provide valuable information? For example, 16S v3/v4 OTUs are ~1000bps but can still be used with high confidence.

- Line 202: Most predicted taxa are Caudoviricetes. Can this be due to a reference database bias in the prediction tools?

- Add a figure summarizing datasets contributing to AVrC. For example proportions of healthy/unhealthy and adult/infant samples.

- Consider performing an analysis of overlaps between the new infant datasets and existing gut viral catalogues.

- Not needed to be addressed, I'm interested in seeing how much overlap there are in vOTUs between those in AVrC (gut microbiome) and other biomes in IMG/VR.

Discussion

- The workflow presented was able to identify significant more vOTUS than other databses described in the introduction. i.e. Gregory et al analyszed 2.7K metagenomes with only 33K species vOTUS. Tisza et al. had 6K metagenomes for 45K species vOTU. With the infant dataset, you identified around ~600K vOTUs from nearly 8K samples. Briefly comment on the key differences in methodology that led to this increase in vOTUs identified.

- Looking at Fig 2, >90% of your sequences were low-quality/not-determined. Then in Fig 3, ~50% of vOTUS are low-quality/not-determined. Briefly discuss the impact of using these low-quality sequences for downstream analysis.

- Line 255: Quantify "high prevalence" or rephrase. Most databases show minor plasmid contamination (<2%), with GVD being the highest at ~10%.

-Line 260: Briefly summarize what is suggested by the benchmark of these detection approaches, and how it's relevent to your case of most of viral sequences are unique to 1 catalogue?

-Line 284: The comparison between AVrC and UHGV needs to be expanded on. For example, the underlying datasets are very similar between UHGV and AVrC. See: https://github.com/snayfach/UHGV#data-sources. Highlight your strength and provide examples in terms of modularity, comprehensiveness, and usability that UHGV lacks.

-Touch on some ecological implications of what you noted in the gut virome, such as the dominance of Caudoviricetes and Spirochaetota-infecting phages, and what this may reveal about gut virome-host dynamics.

- Through out the discussion, there were mentions of bias in datasets/tools. How would you address those bias in future iteration of AVrC?

Figures

- Revise color themes for clarity, use distinct palettes for different plots within the same figure. Ensure semantic consistency between labels and colors. For example, in Fig 1A. Quality and Plasmid should using different color palettes. A 'complete' color in quality have a 'FALSE' label in plasmid, which conflicts on a semantic level.

Methods

- Lines 304-305: Touch upon why were different assemblers used? Assembler choices does have an impact on the OTU prediction.

- Include a workflow diagram summarizing steps for viral sequence identification, clustering, and annotation.

Supplementary Materials

- Supplement File 1: Some database references (e.g., MGV) are missing.

Web Interface

- The following could be due to shinyapps.io hosting issue for large datasets. Consider finding an alternative host that's more reliable:

* Download button on the side bar does not function.

* The filtering doesn't work, unchecking boxes then pressing search doesn't change the table.

- Align bullet margins under the "About/Viral Metadata Description" section.

- Use more human-friendly column headers in the table display.

- Consider adding a GIF/video tutorial for using the website.

Analysis code:

- Consider releasing the analyses code used throughout the manuscript.

**Have the authors made all data and (if applicable) computational code underlying the findings in their manuscript fully available?**

Reviewer #1: Yes

Reviewer #2: **No: ** Datasets used and the SQL table had been released on Zenodo. The Code had been released for the web interface however code used for analyses mentioned throughout the manuscript is missing.

PLOS authors have the option to publish the peer review history of their article (what does this mean? ). If published, this will include your full peer review and any attached files.

**Do you want your identity to be public for this peer review?** For information about this choice, including consent withdrawal, please see our Privacy Policy .

Reviewer #1: **Yes: ** Diego Armando Esquivel Hernández

Reviewer #2: No

**Figure resubmission:**
---

## [Decision Letter · Decision Letter 1]

21 Apr 2025

Dear Mrs Ponsero,

We are pleased to inform you that your manuscript 'The Aggregated Gut Viral Catalogue (AVrC): A Unified Resource for Exploring the Viral Diversity of the Human Gut' has been provisionally accepted for publication in PLOS Computational Biology.

Best regards,

Iddo Friedberg, Ph.D.

Academic Editor

PLOS Computational Biology

Rob De Boer

Section Editor

PLOS Computational Biology

Reviewer's Responses to Questions

**Comments to the Authors:**

Reviewer #1: The authors have improved their manuscript and addressed all reviewer comments and suggestions. Now we need to prepare a very profound proofreading of all the text, but it actually can be accepted for publication at PLOS Computational Biology

Reviewer #2: Dear authors, thank you for addressing my comments and feedback in your revised manuscript. I appreciate the consideration and the detailed responses you provided. The revisions have significantly strengthened the clarity and rigor of your work. Best wishes with your with the next steps in the publication process.

**Have the authors made all data and (if applicable) computational code underlying the findings in their manuscript fully available?**

Reviewer #1: Yes

Reviewer #2: Yes

PLOS authors have the option to publish the peer review history of their article (what does this mean? ). If published, this will include your full peer review and any attached files.

**Do you want your identity to be public for this peer review?** For information about this choice, including consent withdrawal, please see our Privacy Policy .

Reviewer #1: No

Reviewer #2: No

---

## [Editor Report · Acceptance letter]

PCOMPBIOL-D-24-01049R1

The Aggregated Gut Viral Catalogue (AVrC): A Unified Resource for Exploring the Viral Diversity of the Human Gut

Dear Dr Ponsero,

I am pleased to inform you that your manuscript has been formally accepted for publication in PLOS Computational Biology. Your manuscript is now with our production department and you will be notified of the publication date in due course.

With kind regards,

Anita Estes
